# Comparison of a Chromogenic Urine Culture Plate System (UTid+) and Conventional Urine Culture for Canine and Feline Specimens

**DOI:** 10.3390/vetsci9030138

**Published:** 2022-03-16

**Authors:** Stephen D. Cole, Maya Swiderski, Jaclyn Dietrich, Kathryn M. McGonigle

**Affiliations:** 1Clinical Microbiology Laboratory, Department of Pathobiology, School of Veterinary Medicine, University of Pennsylvania, Philadelphia, PA 19104, USA; smaya@sas.upenn.edu (M.S.); jaclyndi@vet.upenn.edu (J.D.); 2Section of Internal Medicine, Department of Clinical Studies and Advanced Medicine, School of Veterinary Medicine, Philadelphia, PA 19104, USA; mcgkat@vet.upenn.edu

**Keywords:** urinary tract infection, bacteriuria, dog, cat, point-of-care diagnostics

## Abstract

In companion animal medicine, urinary tract infection (UTI) is one of the most common indications for antimicrobial therapy. Definitive diagnosis of UTI requires isolation of bacteria with routine urine culture from an animal with concurrent clinical signs. Urine culture is typically performed at reference laboratories where paired susceptibility testing can be performed, but delays in shipment or processing can affect results. This study evaluated the use of a selective chromogenic, point-of-care culture system (UTid+) compared to conventional urine culture. A total of 119 (73 canine and 46 feline) cystocentesis urine samples were evaluated. Conventional urine culture was positive for 28 (23.5%) of the 119 cultures and UTid+ culture was positive for 26 (21.8%). The overall sensitivity, specificity, positive predictive value, negative predictive value and accuracy were 92.3%, 97.8%, 92.3%, 97.8 and 96.6% for UTid+ respectively. Overall, the UTid+ culture system showed an acceptable level of accuracy when compared to conventional urine culture. Agreement of identification results was high (κ = 0.90) with an important exception being *Proteus* spp. which was only identified in 1/3 positive cultures. UTid+ may be useful in scenarios where a common UTI pathogen is expected and identification within 24 h is ideal; however, conventional urine culture remains the gold standard.

## 1. Introduction

Bacterial urinary tract infection (UTI) commonly affects canine and, less commonly, feline patients [1]. UTI is one of the most common reasons for antimicrobial therapy in companion animals [2]. Proper and timely diagnosis of UTI is often crucial for successful treatment and may help to limit antimicrobial administration in cases where it is not indicated [1,2]. Clinical signs of UTI may include dysuria, stranguria, pollakiuria, and/or increased urgency of urination [2]. These clinical signs are indistinguishable from those of other lower urinary tract diseases, including urolithiasis and neoplasia [1]. Definitive diagnosis of UTI requires isolation of bacteria by culture of a sterile urine sample collected by cystocentesis from an animal with concurrent clinical signs [2]. Presence of bacteria in the urine of animals without clinical signs is considered subclinical bacteriuria (SBU). SBU is not considered an indication for antimicrobial intervention in either dogs or cats given that studies have shown animals rarely progress to show clinical signs or to develop deleterious sequelae [3,4]. This recommendation is also consistent with recommendations in human patients [2].

Most veterinarians submit urine culture specimens to reference laboratories; however, delays in shipment or reception may impact results. For example, spiked specimen studies have shown that bacterial concentrations may be artificially decreased if stored at room temperature instead of 4 °C or for longer than 24 h [5,6]. Additionally, extended storage of specimens, particularly those collected by midstream free catch, may lead to overgrowth of clinically insignificant bacterial contaminants which may affect interpretation [7,8]. Anecdotally, submission of urine culture specimens can be delayed by variables such as staffing, patient behavior, owner finances and shipping schedules. Point-of-care (POC) testing for bacteriuria may be desirable in clinical scenarios where submission delays may affect accuracy. Sediment analysis, as part of urinalysis, may be used to identify bacteriuria at POC but interpretation likely depends on the skill of the evaluator and has been shown to have low sensitivity [9]. Other POC tests such as a rapid immunoassay, a urine catalase test, a urine dipstick paddle culture system, and compartmented bacterial culture plates have been evaluated for use in companion animal veterinary medicine and demonstrate variable accuracy [10,11,12,13,14].

This study evaluated a selective, chromogenic culture system for identification of bacterial pathogens associated with UTI. Selective, chromogenic media were used in diagnostic labs for the presumptive identification of bacteria based on growth inhibition of non-target organisms and the production of bacterial enzymes for target organisms that lead to specific colony colors. The system was originally developed for milk culture (Accumast), and uses a single triplate containing three selective chromogenic media [15]. We aimed to compare the diagnostic performance of chromogenic triplates (UTid+) to gold standard conventional urine culture on canine and feline clinical urine specimens.

## 2. Materials and Methods

### 2.1. Specimens and Animal Population

Canine and feline urine specimens were included in this study if a sufficient sample volume (>10 μL) had been submitted to the institutional clinical microbiology laboratory. All samples were submitted as part of the diagnostic work-up of clinical patients as deemed necessary by the attending veterinarian. Samples submitted to the laboratory were collected by veterinarians, licensed veterinary technicians, or veterinary students under the supervision of the former. Specimens were excluded if an insufficient amount of urine was submitted for both culture methods to be performed, or if stored inappropriately (>12 h or non-refrigerated specimens). Specimens were also excluded if collected by a method other than cystocentesis (e.g., free catch, catheterization) due to insufficient surface area on UTid+ plates to enumerate colonies and differentiate between potential contamination during the collection process [2]. Patient species and sex/reproductive status were recorded for each specimen.

### 2.2. Conventional Urine Cultures

All samples were submitted to a university clinical microbiology laboratory for routine aerobic urine culture processed within 1 h of submission. To be included in this study, specimens must have been stored at 4 °C prior to submission for no more than 12 h. Upon receipt, 1 μL of urine was streaked with a calibrated loop in a quantitative fashion onto manufacturer-prepared solid media including tryptic soy agar with 5% sheep’s blood agar (SBA), MacConkey agar and Columbia with colistin-nalidixic acid agar (Remel, Lenexa, KS, USA). All media were stored per manufacturer’s instruction at 4 °C and allowed to come to room temperature prior to plating. Streaked plates were incubated overnight (16–24 h) and examined for growth. Cultures were considered positive if growth of >1000 colony forming unit/mL (CFU/mL) was detected. Each unique morphology for conventional culture was subcultured for purity and identified using the FDA-approved, automated Vitek 2 system with Gram-positive (GP) and Gram-negative (GN) cards (Biomerieux, Marcy L’Atoile, France). If Vitek 2 could not identify an organism then the isolate was sent to a reference laboratory for MALDI-TOF identification (Bruker, Billerica, MA, USA). All work done on the conventional culture was performed by trained clinical microbiology technicians in accordance with laboratory standard operating procedures [16]. Since UTid+ only provides genus level identification for organisms (except for *Escherichia coli*), further comparisons were performed for genus level identification only.

### 2.3. UTid+ Culture

In parallel, urine from the same specimen was streaked using a 1 μL calibrated loop onto each section of the triplate (UTid+, FERA Diagnostics and Biologicals, College Station, TX, USA). Triplates were incubated for 20–24 h at 35 °C. The individual interpreting the UTid+ result was blinded to the conventional culture result. UTid+ plates were interpreted as positive if any growth was present. UTid+ plates with no growth were interpreted as negative. Bacterial identification was interpreted using the manufacturer’s instructions based on which media showed colony growth and the colonies’ color. The urine triplates contain three distinct media that are able to grow and differentiate the following bacteria: (Section 1) *Klebsiella* spp., *E. coli*, *Pseudomonas* spp., *Proteus* spp., (Section 2) *Streptococcus* spp., *Enterococcus* spp., and (Section 3) *Staphylococcus* spp. [15]. Results (positive vs. negative, bacterial identification) for both routine and triplate cultures were recorded in an excel spreadsheet.

### 2.4. Statistical Analysis

Sensitivity, specificity, positive predictive value (PPV), and negative predictive value (NPV) for the UTid+ plates were calculated based on true positives, true negatives, false positives and false negatives, comparing results to conventional urine culture results. In addition, accuracy was calculated by dividing the number of true positives and true negatives by the total number of tests. The simple Cohen’s kappa coefficient (κ) was calculated using the FREQ procedure, which generates frequency and crosstabulation tables, in SAS version 9.4 (SAS/STAT, SAS Institute Inc., Cary, NC, USA). This parameter assumed that the two response variables (UTid+ and gold standard culture) were independent ratings, and the coefficient equaled 1 when there was complete agreement between the two tests. The null hypothesis for this test was that if agreement happened due to chance the Kappa coefficient was equal to zero. Under this null hypothesis, *p* values associated with this test equal or smaller than 0.05 were considered significant. The aforementioned analyses were repeated independently according to host species (dog and cat).

### 2.5. Review of Antimicrobial Prescribing for Patients with Discordant Results

To assess the potential clinical impact of UTiD+ errors, a review of medical records was performed for animals with discordant UTiD+ and conventional urine culture results. Steps taken by clinicians based on conventional urine culture results were characterized as either (1) prescription of a new antimicrobial, (2) continuation of a current antimicrobial, (3) discontinuation of antimicrobial therapy or (4) no therapy initiated. The actual steps taken were then directly compared to hypothetical steps taken following UTid+ results, based on International Society for Companion Animal Infectious Disease (ISCAID) guidelines and published rates of susceptibility for specific organisms [2].

## 3. Results

### 3.1. Specimens and Animal Population

In total, 119 (73 canine and 46 feline) cystocentesis urine specimens were tested by both UTid+ and conventional culture methods. The canine specimens were from 4 intact females, 23 castrated males and 46 spayed females. The feline specimens were from 24 castrated males and 22 spayed females.

### 3.2. Conventional Urine Culture

Conventional urine culture was positive for 28 (28/119, 23.5%) of the 119 specimens. A single organism was isolated from 21 of the positive conventional cultures and two distinct organisms from seven of the positive cultures. Figure 1 illustrates the prevalence of pathogens isolated by conventional urine culture compared with UTid+ identification results. The most common pathogens identified by automated biochemical testing were *E. coli* (*n* = 12, 42.8% of positive cultures) followed by *Enterococcus* spp. (*n* = 9, 32.1% of positive cultures) and *Staphylococcus* spp. (*n* = 8, 28.5% of positive cultures). Proportionally less of the feline conventional urine cultures were positive (8/46, 17.3%) than the canine conventional urine cultures (20/73, 27.4%).

### 3.3. UTid+ Culture

The UTid+ plates identified 26 positive urine samples (21.8%). A single organism was identified from 19 specimens, two distinct organisms from six specimens, and three distinct organisms from one specimen. The most common pathogens presumptively identified by UTid+ were *E. coli (**n* = 12, 43.7% of positive cultures), *Enterococcus* spp. (*n* = 10, 35.7% of positive cultures) and *Staphylococcus* spp. (*n* = 8, 30.7% of positive cultures). Overall, a total of six (6/119, 5.0%) UTid+ cultures did not match the results obtained from conventional urine culture. The UTid+ had a false negative for two cultures (2/119, 1.6%), both of which grew *Proteus mirabilis* on conventional urine culture. Misidentification occurred for one organism (*Corynebacterium auriscanis* identified via MALDI-TOF was identified a *Staphylococcus* spp. by UTid+). One UTid+ presumptively identified an additional organism (*Enterococcus* spp.) in addition to the two (*E. coli* and *Streptococcus* spp.) found by conventional culture in one specimen and in another specimen (*n* = 1). UTid+ could not distinguish between two morphologically different staphylococci identified in conventional urine culture. Finally, UTid+ failed to identify one (*Streptococcus* spp.) of two organisms in a single culture.

### 3.4. Statistical Analysis

The overall sensitivity, specificity, PPV, NPV, accuracy and κ coefficient were 92.3%, 97.8%, 92.3%, 97.8%, 96.6% and 0.90 (*p* < 0.001) for UTid+ when compared with conventional urine culture. Table 1 illustrates these results as well as test characteristics according to animal species.

### 3.5. Review of Antimicrobial Prescribing for Patients with Discordant Results

A new beta-lactam antimicrobial (amoxicillin or amoxicillin-clavulanate) was prescribed based on susceptibility testing results for 5/6 animals with discordant results. The sixth case (misidentification of *C. auriscanis* as *Staphylococcus* spp.) was prescribed a new course of marbofloxacin. In 3/6 cases, the beta-lactam antibiotic would have been an appropriate empiric drug for the organisms identified by both UTId+ and conventional urine culture [2,17]. For the two false negative cases, UTid+ results would have suggested that antimicrobials should not have been prescribed; however, based on conventional urine culture, antimicrobials were prescribed to treat *Proteus* spp. [2]. For the misidentification case of a *Staphylococcus* spp., an ISCAID first tier UTI antimicrobial (amoxicillin, amoxicillin-clavulanate or trimethoprim-sulfamethoxazole) should hypothetically have been prescribed [2,18]. These first-line antimicrobials would not be a good empiric decision, based on published susceptibility rates (<50%) of *C. auriscanis* from dermatitis cases; however, this should be interpreted with caution given that urine isolate resistance rates for *C. auriscanis* are not currently found in the literature [19]. Alternatively, the isolate could have been sent to a clinical microbiology laboratory where this issue may have been resolved easily.

## 4. Discussion

POC tests for bacteriuria are desirable tools as they do not require transport to a laboratory, which may decrease turn-around-time and increase costs. The ideal POC would be a highly sensitive test which can be used to rule out the presence of bacteria in the urine for clinical scenarios such as prior to urogenital tract surgery or in cases of feline lower urinary tract disease (FLUTD) [1,2] This ideal test would also be highly specific and could be used to confirm clinical diagnosis of UTI where proper identification of a pathogen could provide additional empiric information on which antimicrobial therapy to choose [1,2]. Overall, the UTid+ culture system showed an acceptable level of accuracy (96.6%) when compared to conventional urine culture. It was also a relatively sensitive (92.3%) and specific (97.8%) test. Agreement of results was almost perfect with a kappa coefficient of 0.90 (*p* < 0.001) which denotes near-perfect agreement between the tests.

In our study, UTid+ performed comparably to other POC tests previously evaluated in the veterinary literature. Ybarra et al. compared conventional urine culture with a urine dipstick paddle system and found a slightly higher sensitivity and specificity than UTid+ (97.3% and 98.6% respectively), but lower accuracy for identification of bacteria (75.8%) [12]. Olin et al. evaluated a culture-based POC test which had similar overall accuracy (94%), but lower sensitivity (81.0%) and higher specificity (99%) than UTid+ [13]. UTid+, in terms of sensitivity and specificity, performed considerably better than results published for a catalase-based urine test (89.0% and 71.0% respectively), but worse than results for a rapid immunoassay (RIA-97.4% and 98.8% respectively) [10,14]. Neither of these can identify bacteria to a genus level like UTid+, but the RIA can distinguish between Gram-negative and Gram-positive bacteria [10].

Hypothetically, if UTid+ results had been used to guide antimicrobial therapy for animals in this study, there would have been three instances of possible misprescribing (3/119, 2.5%). One case was due to misidentification and there were two cases of false negatives. The misidentification was of *C. auriscanis* as *Staphylococcus* spp. The clinical relevance of corynebacteria in the urinary tracts of companion animals has not been well studied due to their low prevalence [20,21]. It is unclear in this case if antimicrobial therapy was even warranted and, in general, it is unlikely that this misidentification would be a common problem in clinical practice.

Even though the UTid+ had a low false negative rate (1.6%), the UTid+ system only isolated *Proteus* spp. in 1/3 cultures. *Proteus* spp. is commonly associated with struvite urolithiasis and these results may suggest that UTid+ should not be used in lieu of conventional culture in cases of animals with crystalluria or bladder stones [22]. *Proteus* spp. readily grows on standard laboratory media (e.g., TSA agar, MacConkey agar) at the temperature and atmospheric conditions used in this study [23]. The composition of the UTid+ plates is proprietary so it is difficult to determine if any media composition limitations may decrease sensitivity to *Proteus* spp. A study of the same chromogenic media in cattle with mastitis found no animals with *Proteus* spp. in its study population [15]. In a study of a different POC urine test, one of three *Proteus* spp.-positive cultures showed the same type of discordant result as the UTid+ system (positive on conventional culture and negative on POC test) but the non-culture-based assays detected *Proteus* spp. completely [10,13,14].

In our study, we found an overall bacteriuria prevalence of 23.5% in the submitted urine specimens by the gold standard (conventional urine culture). The PPV and NPV were 92.3% and 97.8% respectively; however, PPV and NPV of diagnostic tests are dependent on the prevalence of a disease in the tested population. PPV and NPV have an inverse relationship; as prevalence increases, the PPV of a test also increases and the NPV decreases, and as prevalence decreases, then PPV decreases but NPV increases [24]. In our study, this relationship is demonstrated by performance of the UTid+ plates on specimens from different animal species. While sensitivity and specificity results were fairly similar (>90%) for both dogs and cats, the PPV for cats (75.0%) was much lower than for dogs (100.0%). The lower PPV in our study was influenced by the lower prevalence of bacteriuria in cats (17.3%) than dogs (27.4%) based on conventional urine culture. This difference between dogs and cats is well documented in the veterinary literature [1,2,3]. Clinicians who choose to use UTid+ in their patient population should remember that PPV and NPV results are best applied in a similar clinical population to that of our study (e.g., animals at a referral hospital, animal species), or consider how their populations may differ when interpreting results.

The results of this study did not evaluate urine collected by free-catch or catheter collection methods. The smaller surface area of the UTid+ triplates makes accurate colony counts difficult to achieve, which may limit their utility in certain cases [1,2,25]. Because clinical indications for urine culture were not assessed in this study, it is important to note that this study did not limit evaluation to include only animals with clinical signs of UTI. In addition to animals with UTI, it is likely that culture testing was performed for some animals because of other clinical indications such as SBU characterization or as pre-surgical screening for urogenital surgery patients. There are growing diagnostic stewardship efforts in human medicine which promote less frequent use of urine culture since a positive result will often trigger therapeutic intervention which may not be needed in a patient without clinical signs [26]. It is likely true that similar phenomena take place in veterinary medicine, and practices such as reflex urine culture based on urinalysis results should be limited to animals with clinical signs [2,27,28].

While POC testing has the advantages of faster turn-around-time and lower costs, it also has several distinct disadvantages. First and foremost, in most situations you cannot pair positive results with immediate antimicrobial susceptibility testing, which ultimately may delay these results. For culture-independent approaches the ability to identify bacterial genera or species is also limited. Culture of uropathogens within a veterinary hospital environment is not without biosafety risks [29]. A survey of veterinarians performing in-house urine culture identified widespread unsafe practices such as inappropriate disposal, lack of biosafety protocols and inadequate laboratory spaces separated from clinical care areas [29]. It is also important to point out that shipment of cultured bacteria on media may be distinctly different from transport of exempt animal specimens (i.e., urine without a high likelihood of containing a pathogen). Shipping of bacterial cultures is subject to national (e.g., Department of Transportation in USA) and international (e.g., International Air Transport Association) regulations and should be performed by trained individuals [29]. Most uropathogens are characterized as Category B, and additional requirements often include specific packaging and labeling [30]. Veterinarians who choose to perform POC culture testing should make sure all appropriate biosafety measures are in place to do so.

Diagnostic tests may perform differently if limited to a particular population, for example, only those with clinical signs of UTI [10,11]. UTid+, as with all diagnostic tests, should always be interpreted in the context of the clinical patient and should only form part of the clinical decision-making process. Future studies of UTid+ should include prospective evaluation in clinical populations where this may prove to be valuable tool, such as those with upper vs. lower urinary tract infections, or be restricted to a population where UTI is likely based on clinical history.

## 5. Conclusions

The UTid+ system should not be considered a replacement for conventional bacterial culture and susceptibility. UTid+, when paired with cystocentesis specimens from dogs and cats, had an overall test accuracy of 96.6%. However, UTid+ can not be paired with susceptibility testing unless culture is sent to a reference laboratory. The test also cannot be used to enumerate bacterial concentrations accurately, which is an important step in interpreting clinical relevance of results for catheter or clean free-catch specimens. In addition, UTid+ should not be used alone for cases where there is a high index of suspicion for *Proteus* spp. bacteriuria. Before implementation of UTid+, clinics will need to invest in a reliable incubator and to develop specific biosafety protocols. The UTid+ system offers several distinct advantages when compared to conventional urine culture, including a potentially lower cost option for pet owners and relatively rapid turn-around time without shipment (although overnight incubation is still required). The UTid+ system should be viewed as an additional tool to aid in the clinical decision-making process surrounding bacteriuria and be used to optimize antimicrobial and diagnostic stewardship in veterinary practices.

## Figures and Tables

**Figure 1 vetsci-09-00138-f001:**
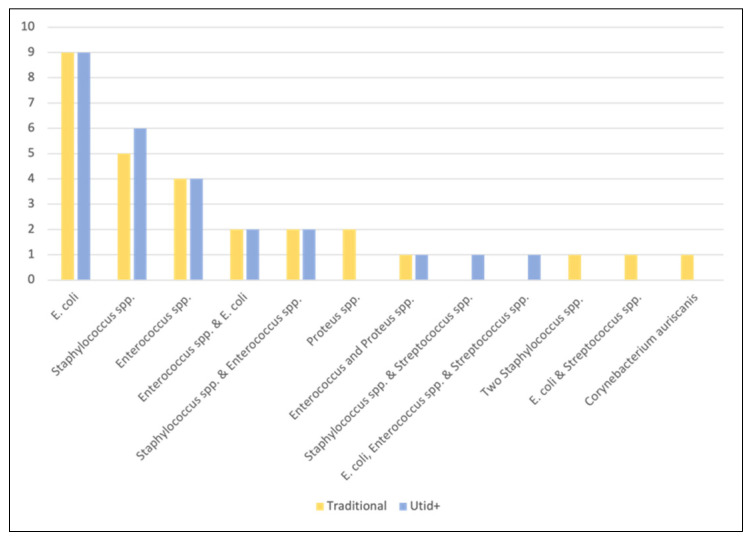
Distribution of bacteria identified from urine specimens by method.

**Table 1 vetsci-09-00138-t001:** Diagnostic characteristics of UTId+ plates compared to conventional urine culture for specimens collected by cystocentesis for dogs and cats presenting to a referral veterinary hospital.

	Overall	Canine Specimens	Feline Specimens
Specimen Total	119	73	46
Sensitivity %	92.3 (75.9–97.9)	100 (93.2–100)	95.0 (83.5–98.6)
Specificity, %	97.8 (92.5–99.4)	90.0 (69.9–97.2)	100 (61.0–100)
Positive Predictive Value, %	92.3 (75.9–97.9)	100 (82.4–100)	75.0 (40.9–92.9)
Negative Predictive Value, %	97.8 (92.5–99.4)	96.4 (87.7–99.0)	100 (90.8–100)
Accuracy, %	96.6 (91.7–98.7)	97.2 (90.4–99.6)	95.6 (85.1–99.4)
κ ^1^, %	0.90 (0.80–0.99)	92.8 (83.2–100)	83.2 (60.7–100)
κ *p*-value	<0.001	<0.001	<0.001

^1^ Cohen’s kappa coefficient. κ = 0 denotes poor agreement; 0.01 to 0.20 denotes slight agreement; 0.21 to 0.40 denotes fair agreement; 0.41 to 0.60 denotes moderate agreement; 0.61 to 0.80 denotes substantial agreement and 0.81 to 1.00 denotes almost perfect agreement.

## Data Availability

Raw data will be made upon request via email at scole@vet.upenn.edu.

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
