# Peer review of "Comparison of a Chromogenic Urine Culture Plate System (UTid+) and Conventional Urine Culture for Canine and Feline Specimens"

_vetsci, 2022, doi:10.3390/vetsci9030138_

Round 1
Reviewer 1 Report
In this manuscript, the authors describe the comparison of a point-of-care test regarding bacteriuria diagnostic performance to the gold standard method, namely the urine culture.
The manuscript is well-written and organized. The methodology along with the results of the research are presented explicitly. The concept and the findings of this study should hold the attention of small animal practitioners as well veterinary microbiologists.
Nonetheless, some minor issues should be addressed, as follows:
Line 34: parentheses are used for the reference (1) instead of brackets [ ], probably due to mistyping.
Line 36: parentheses are also used for the reference (2).
Line 45: The authors state that "bacterial concentrantions may be artificially decreased if stored at room temperature or for longer than 24 hours". The temperature of storage (e.g. 4oC - refrigeration) is omitted and should be added.
Line 186, 195: An additional space ( ) is typed.
Line 199: The words "evaluate" and "catheter" are misspelled, so please rephrase.
Line 224: The authors state that "... shipment of bacterial cultures is distinctly different then transport of exempt animal specimens ...". Maybe the word is then instead of than? If this is the case, please rephrase.
General comments
1)In this study, the references cited by the authors are relevant and up-to-date. However, a more thorough literature review should be conducted. 15 out of 19 references are cited in the Introduction section whereas only three new references are cited in the Discussion section, and only two of them refer to diagnostic methods regarding animal specimens. The findings of other similar studies should be commented in the Discussion section. Apart from that, references could be added to: a) Lines 175-180 - concerning the ideal point-of-care diagnostic test, b) Lines 186-189 - explaining why Proteus spp. cannot be detected by this or other point-of-care methods during similar studies (if any), as in the study provided as reference, conventional culture has been used, c) lines 209-210 "similar phenomena .... as needed" d) lines 217-220 e) lines 221-223 - concerning biohazards deriving from uropathogens in hospital environment.
2)Line 223-225: "It is also important ... training and approvals". To the best of my knowledge, shipment/transport of bacterial cultures within a hospital/clinic or to a specialized laboratory is an established and frequent procedure, if certain biosecurity conditions/measures are properly met. Moreover, appropriate packing and shipment of specimens should not require too specialized training, especially for veterinarians or experienced veterinary technicians. Of course, it is unambiguous that point-of-care testing is more convenient, as no transport/shipment is needed. May the authors explain this point in a more clear way and provide a reference?
c) Lines 225-228 and Conclusions are identical. Consequently, the authors should rephrase the Conclusions section. May the authors stress the advantages of using the UTid+ test (e.g. it is a low-cot alternative for pet-owners, etc) along with its diagnostic performance and perspectives for the future.
Author Response
Thank you for your insightful edits and comments.
AU: We have addressed all of the highlighted minor issues in the manuscript.
Line 34: parentheses are used for the reference (1) instead of brackets [ ], probably due to mistyping.
Line 36: parentheses are also used for the reference (2).
Line 45: The authors state that "bacterial concentrantions may be artificially decreased if stored at room temperature or for longer than 24 hours". The temperature of storage (e.g. 4oC - refrigeration) is omitted and should be added.
Line 186, 195: An additional space ( ) is typed.
Line 199: The words "evaluate" and "catheter" are misspelled, so please rephrase.
Line 224: The authors state that "... shipment of bacterial cultures is distinctly different then transport of exempt animal specimens ...". Maybe the word is then instead of than? If this is the case, please rephrase.
General Comments
1. In this study, the references cited by the authors are relevant and up-to-date. However, a more thorough literature review should be conducted. 15 out of 19 references are cited in the Introduction section whereas only three new references are cited in the Discussion section, and only two of them refer to diagnostic methods regarding animal specimens. The findings of other similar studies should be commented in the Discussion section. Apart from that, references could be added to: a) Lines 175-180 - concerning the ideal point-of-care diagnostic test, b) Lines 186-189 - explaining why Proteus spp. cannot be detected by this or other point-of-care methods during similar studies (if any), as in the study provided as reference, conventional culture has been used, c) lines 209-210 "similar phenomena .... as needed" d) lines 217-220 e) lines 221-223 - concerning biohazards deriving from uropathogens in hospital environment.
AU: Thank you for this! We believe this has greatly improved our discussion section and helped to lengthen the manuscript per journal requirements.We have added an additional 15 references and expanded our discussion on all of these points.
2)Line 223-225: "It is also important ... training and approvals". To the best of my knowledge, shipment/transport of bacterial cultures within a hospital/clinic or to a specialized laboratory is an established and frequent procedure, if certain biosecurity conditions/measures are properly met. Moreover, appropriate packing and shipment of specimens should not require too specialized training, especially for veterinarians or experienced veterinary technicians. Of course, it is unambiguous that point-of-care testing is more convenient, as no transport/shipment is needed. May the authors explain this point in a more clear way and provide a reference?
AU: Thank you for this! This section was initially too vague and has now been revised to better articualte the concerns with shipping. WHile we agree that vets/vet techs are more than capable, we would like to highlight that these requirements in case veterinarians are unaware of them.
c) Lines 225-228 and Conclusions are identical. Consequently, the authors should rephrase the Conclusions section. May the authors stress the advantages of using the UTid+ test (e.g. it is a low-cot alternative for pet-owners, etc) along with its diagnostic performance and perspectives for the future.
AU: Thank you! This was mainly an oversight during formatting into the Vet Sciences template. We have corrected this and expanded our conclusion.
Reviewer 2 Report
Dear Authors,
I appreciate the item of POC in UTI and I believe it is an issue must be looked at in greater depth.
I have just few minor point:
- In Materials and Methods I would appreciate a more detailed description of the method UTI+ studied;
- Results in my opinion should be discussed also in relation to the clinical decision. It could happen, for example in the case of 2 or more pathogens, that result of UTI+ differs from gold standard but both result lead to the same clinical prescription. Indeed authors state that future studies of UTI+ should include prospective evaluation in clinical populations, I would include results in clinical population also in the present study.
Author Response
Thank you for your comments and review. We have addressed your comments below:
- Materials and Methods I would appreciate a more detailed description of the method UTI+ studied
AU: Thank you for this comment. We have further described the interpretation of the UTId+ plates at line 112-115.
- Results in my opinion should be discussed also in relation to the clinical decision. It could happen, for example in the case of 2 or more pathogens, that result of UTI+ differs from gold standard but both result lead to the same clinical prescription. Indeed authors state that future studies of UTI+ should include prospective evaluation in clinical populations, I would include results in clinical population also in the present study
AU: Thank you for this comment. We have completed medical record review and addressed this is sections 2.5 and 3.5. It is also discussed in lines 235-242 of the discussion.